# Impact of *Chlorella vulgaris* Intake Levels on Performance Parameters and Blood Health Markers in Broiler Chickens

**DOI:** 10.3390/vetsci11070290

**Published:** 2024-06-28

**Authors:** Ana R. Mendes, Maria P. Spínola, Madalena Lordelo, José A. M. Prates

**Affiliations:** 1LEAF—Linking Landscape, Environment, Agriculture and Food Research Center, Associate Laboratory TERRA, Instituto Superior de Agronomia, Universidade de Lisboa, Tapada da Ajuda, 1349-017 Lisboa, Portugal; rmendes@isa.ulisboa.pt (A.R.M.); mlordelo@isa.ulisboa.pt (M.L.); 2CIISA—Centro de Investigação Interdisciplinar em Sanidade Animal, Faculdade de Medicina Veterinária, Universidade de Lisboa, Av. da Universidade Técnica, 1300-477 Lisboa, Portugal; mariaspinola@fmv.ulisboa.pt; 3Associate Laboratory for Animal and Veterinary Sciences (AL4AnimalS), Av. da Universidade Técnica, 1300-477 Lisboa, Portugal

**Keywords:** microalgae, poultry, cumulative intake, growth performance, health marker, plasma metabolite

## Abstract

**Simple Summary:**

This review explores how the *Chlorella vulgaris* microalga affects broiler chickens, specifically their growth performance and blood health markers. Our analysis shows that a cumulative intake of approximately 20 g per bird improved growth and feed efficiency, with benefits peaking at this level before levelling off. This amount also enhanced plasma health markers, suggesting that *C. vulgaris* can support overall broiler health. However, exceeding 20 g diminished these benefits, emphasising the need to determine the optimal intake levels of *C. vulgaris*.

**Abstract:**

This systematic review examines the effects of cumulative *Chlorella vulgaris* intake levels on broiler chickens, focusing on growth performance and systemic health markers. The review establishes a clear relationship between cumulative *C. vulgaris* intake and significant outcomes in poultry nutrition and health through a detailed analysis of various studies. The correlation analysis revealed that cumulative *C. vulgaris* intake levels ranging from 0.8 to 718 g/bird influenced growth rates and feed efficiency, following sigmoid models. Specifically, intakes of approximately 20 g/bird maximized final body weight (R^2^ = 0.616, *p* < 0.001), cumulative body weight gain (R^2^ = 0.627, *p* < 0.001) and daily weight gain (R^2^ = 0.639, *p* < 0.001). The feed conversion ratio also improved with increasing *C. vulgaris* intakes up to this level, although this was non-significant (R^2^ = 0.289, *p* = 0.117). In addition, similar cumulative *C. vulgaris* intake levels impacted plasma health markers in broilers, leading to reductions in triacylglycerols and cholesterol and improvements in immunoglobulin levels. These findings underscore the importance of carefully calibrated *C. vulgaris* supplementation strategies to optimise poultry growth and health without adverse effects. Future research should focus on refining *C. vulgaris* dosing guidelines and further exploring its long-term effects and mechanisms of action to enhance poultry health and production sustainability.

## 1. Introduction

The world’s population is increasing and is estimated to reach 9.7 billion by 2050 [1]. As such, the search for sustainable and health-enhancing livestock feed has become increasingly urgent, with a growing emphasis on natural additives and feedstocks that improve the well-being and productivity of farm animals [2,3]. In this context, *Chlorella vulgaris*, a protein-rich microalga, has garnered significant attention as a potent feed supplement in poultry diets. Its potential to substantially influence the growth performance and health status of broiler chickens, in addition to its nutritional benefits, is an area of extensive research and interest [4,5].

*C. vulgaris* is valued for its high protein content and presence of vitamins, minerals and essential fatty acids. These nutritional components improve growth rates and enhance immune responses, leading to better overall health in broilers. Studies have shown that incorporating *C. vulgaris* into poultry diets can significantly improve feed conversion ratios (FCR), carcass quality and various health markers [4,6,7]. As such, *C. vulgaris* represents a promising alternative to conventional feed additives, aligning with the growing demand for more sustainable and health-promoting livestock feed options.

The unicellular nature of *C. vulgaris* makes it highly valued for its rich nutritional composition, which includes essential nutrients like proteins, vitamins, minerals and bioactive compounds such as chlorophylls, carotenoids and omega-3 (n−3) fatty acids [8,9,10,11,12]. Additionally, *C. vulgaris* has a favourable essential amino acid composition [13]. These components not only serve as nutritional supplements but also significantly enhance the growth and health of broilers. Chlorophylls, for instance, play crucial roles in detoxification processes, while carotenoids, including beta-carotene, contribute to immune function and visual health. The presence of omega-3 fatty acids in *C. vulgaris* also supports cardiovascular health and reduces inflammation, further promoting the overall well-being of broilers [4]. 

Studies have demonstrated that *C. vulgaris* supplementation leads to marked improvements in broiler growth performance and feed efficiency. *C. vulgaris*’s high protein content and balanced amino acid profile contribute to muscle development and weight gain. For example, the dietary inclusion of *C. vulgaris* improved broilers’ final body weights and FCRs, indicating a more efficient conversion of feed into body mass [14]. Additionally, *C. vulgaris* supplementation has been shown to enhance the nutritional quality of meat by increasing the levels of beneficial fatty acids and reducing harmful lipid oxidation, thereby improving meat quality [3,15,16] and consumer acceptability.

In addition to their substantial protein contents, microalgae are rich in a variety of bioactive compounds, including polysaccharides, polyphenols and pigments [17]. *C. vulgaris*, in particular, demonstrates potent immune-enhancing properties. The bioactive compounds in *C. vulgaris*, such as beta-glucans and other polysaccharides, modulate the immune system, enhancing both innate and adaptive immunity in broilers, leading to increased antibody production, improved disease resistance and overall health [8,11,18,19]. For instance, supplementation with *C. vulgaris* has been associated with increased levels of immunoglobulin IgA, IgM and IgG, which are critical for immune defence. Moreover, the antioxidant properties of *C. vulgaris*’s bioactive compounds [20] are effective in reducing oxidative stress, thereby protecting broiler cells from damage and supporting healthy physiological functions [6,7]. This positions *C. vulgaris* as a valuable addition to broiler diets, contributing to productivity and animal welfare.

However, integrating *C. vulgaris* into broiler diets presents several challenges, particularly concerning the appropriate levels of inclusion, the duration of feeding and cell wall indigestibility for monogastric animals like broilers. These factors are critical in determining the overall impact on growth performance and health. While certain levels of *C. vulgaris* supplementation have shown promise in enhancing growth performance, variations in dosage and feeding duration can produce different outcomes in terms of health benefits and physiological responses. For instance, studies have indicated that low inclusion levels (up to 2% of the diet) can improve FCR without negatively affecting growth, while higher levels might not provide additional benefits or could potentially lead to adverse effects [6,16].

One significant challenge is the cell wall of *C. vulgaris*, which is highly resistant to digestion in monogastrics due to its rigid structure composed of sporopollenin-like biopolymers. This indigestibility can limit the bioavailability of nutrients contained within this microalga, reducing its effectiveness as a feed supplement. Techniques such as mechanical disruption, enzymatic treatment, fermentation or a pulse-electric field are often required to break down its cell walls and enhance its nutrient availability, adding to the complexity and cost of using *C. vulgaris* in broiler diets [4,21,22,23,24].

The main aim of this review was to systematically assess and synthesise existing scientific literature from databases such as Google Scholar (Google LLC, Mountain View, CA, USA), PubMed (NCBI, Bethesda, MD, USA), Scopus (Elsevier B.V., Amsterdam, The Netherlands) and Web of Science (Clarivate Analytics, Philadelphia, PA, USA). The goal was to determine the dose–response relationship between various cumulative levels of *C. vulgaris* intake and its impact on key performance parameters and health markers in broiler chickens. Cumulative microalga intake was calculated by multiplying the total feed consumed by a bird by the proportion of *C. vulgaris* in its diet. We hypothesised that the observed effects resulted from the unique transfer kinetics of *C. vulgaris*’s bioactive compounds to the birds. This review aimed to identify the optimal *C. vulgaris* dosage ranges that maximised growth performance and health benefits in broiler chickens. It also aimed to highlight any potential thresholds or limits beyond which *C. vulgaris* incorporation could lead to diminishing returns or adverse effects on these parameters.

## 2. Impact of Cumulative *Chlorella vulgaris* Intake Levels on the Growth Performance of Broilers

Table 1 summarizes the data from various studies that examined the effects of different cumulative intake levels of *C. vulgaris* on broiler growth performance. An analysis of the nutritional profile of *C. vulgaris* was previously performed [25]. *C. vulgaris*, known for its rich nutritional profile, has been the subject of numerous studies aiming to quantify its benefits on broiler performance [25]. These studies have systematically investigated *C. vulgaris*’s effects, considering variables such as the ages and initial weights of the broilers, the percentage of microalgae in their diets, the durations of the supplementation periods and the cumulative intakes alongside critical growth performance indicators. The cumulative intake levels and their impacts varied across the studies, providing a broad perspective on the effects of this microalga. The data show varying cumulative *C. vulgaris* intake levels, ranging from 0.8 g/bird [26] to 718 g/bird [6], with associated changes in the birds’ growth outcomes.

In one study, the effects of a 1.40 g cumulative *C. vulgaris* intake per bird over 34 days were investigated, observing a final body weight of 1533 g, a cumulative body weight gain of 1488 g, and a daily body weight gain of 43.8 g, with an FCR of 1.88 [27], and all of these effects were not significantly different from the control treatment. In another part of this study, with a cumulative intake of 4.27 g over the same period, the final body weight was 1619 g, with similar body weight gain patterns and an FCR of 1.81 [27]. Another study using a cumulative intake of 14.13 g showed a final body weight of 1643 g and an FCR of 1.78, indicating a slight improvement in feed efficiency with increased *C. vulgaris* levels [27]. In these two studies, the authors presented final body weights and feed conversion ratios that were better than the controls. Another study reported on the effects of a 3.52 g cumulative *C. vulgaris* intake over 31 days, noting a final body weight of 1990 g and a body weight gain of 1916 g, with an FCR of 1.84 [28]. In other studies, higher intake levels of *C. vulgaris*, such as 20.0 g over 41 days, resulted in a final body weight of 2166 g and an FCR of 1.571, with a dressing percentage of 71.69% and a breast-meat water-holding capacity of 88.33% [29]. In contrast, the control treatment achieved a final body weight of 1791 g, with an FCR of 1.784, indicating significant improvements in growth performance at these higher intake levels of *C. vulgaris*. Additionally, studies have examined even higher cumulative *C. vulgaris* intake levels, such as 401 g, 561 g and 718 g over 34 days [6]. They reported final body weights of 2819 g, 2587 g and 2342 g, respectively. The FCRs were 1.5, 1.53 and 1.61, respectively, indicating that while higher *C. vulgaris* intake levels could promote growth, the efficiency gains might diminish at very high inclusion rates.

**Table 1 vetsci-11-00290-t001:** Impact of cumulative *Chlorella vulgaris* intake levels on the growth performance of broilers.

Initial Age and Weight	Alga Level (% Feed) and Duration of Trial (Days) ^1^	Cumulative Alga Intake (g/bird) ^2^	Growth Performance				Reference
			Final Body Weight (g)	Cumulative Body Weight Gain (g)	Body Weight Gain (g/d)	Feed Conversion Ratio	
39.43 g, 1 d-old ^3^	0.50%, 9 d	0.800	190.3	150.9	16.8	1.12	[26]
45.1 g, 1 d-old ^3^	0.05%, 34 d	1.40	1533	1488	43.8	1.88	[27]
1 d-old ^3^	0.07%, 41 d	2.95	-	2723.1	66.4	1.55	[30]
72.56 g, 4 d-old ^3,4^	0.10%, 31 d	3.52	1990	1916	61.8	1.84	[28]
45.1 g, 1 d-old ^3^	0.15%, 34 d	4.27	1619	1574	46.3	1.81	[27]
40.03 g, 1 d-old	0.10%, 41 d	4.35	2501.3	2461.3	60	1.77	[31]
1 d-old ^3^	0.14%, 41 d	5.94	-	2755	67.2	1.54	[30]
41.8 g, 1 d-old	0.20%, 41 d	6.71	2001	1959	47.8	1.713	[29]
40.03 g, 1 d-old	0.20%, 41 d	8.73	2520.8	2480.8	60.5	1.76	[31]
1 d-old ^3^	0.21%, 41 d	9.22	-	2850.8	69.5	1.54	[30]
41.8 g, 1 d-old	0.40%, 41 d	13.0	2077	2035	49.6	1.602	[29]
45.1 g, 1 d-old ^3^	0.50%, 34 d	14.1	1643	1598	47	1.78	[27]
41.8 g, 1 d-old	0.60%, 41 d	20.0	2166	2124	51.8	1.571	[29]
1 d-old ^3^	1.0%, 34 d	24.4	-	1603	47.1	1.52	[32]
1 d-old ^3,5^	1.0%, 34 d	25.2	-	1647	48.4	1.53	[32]
47.1 g, 1 d-old ^3^	0.80%, 34 d	28.9	2606.3	2559	73.1	1.45	[33]
788 g, 21 d-old ^3^	10%, 14 d	176	1928	1140	81.4	1.54	[15]
107 g, 5 d-old ^3^	10%, 34 d	401	2819	2712	77.49	1.5	[6]
109 g, 5 d-old ^3^	15%, 34 d	561	2587	2478	70.8	1.53	[6]
106 g, 5 d-old ^4^	20%, 34 d	718	2342	2236	63.87	1.61	[6]

^1^ The final day of the trial, which involved slaughtering, was not included in the trial’s duration. ^2^ Calculated by multiplying the total feed consumed per animal during the experimental period by the dietary percentage of microalga. For some of the studies, no information about the cumulative feed intake was available, and therefore, an estimation of this was completed as follows: cumulative feed intake (g/bird) (An et al. [27] and Roques et al. [33]) = CFI (g/d/bird) × number of trial days; cumulative feed intake (g/bird) (Alfaia et al. [15]) = cumulative feed intake (g/d/pen) × number of trial days/number of birds; cumulative feed intake (g/bird) (Cabrol et al. [6]) = cumulative feed intake (g/pen)/number of birds; and cumulative feed intake (g/bird) (Rezvani et al. [30]) = cumulative body weight gain (g) × feed conversion ratio. ^3^ Male broilers. ^4^ Female broilers. ^5^ In this trial, fresh liquid *Chlorella vulgaris* was used.

Table 2 summarises the correlation analysis for predicting the dependent performance variables based on cumulative *C. vulgaris* intake. Only variables with three or more degrees of freedom (dof) were analysed for correlation, ensuring the reliability of the statistical analysis. Data analysis for the correlations was conducted using SPSS software (version 29.0, 2024), employing various regression and curve estimation techniques. These included linear, logarithmic, inverse, quadratic, cubic, compound, power, sigmoid, growth, exponential and logistic models. The analysis focused on cumulative *C. vulgaris* intake as the independent variable that influenced the growth performance metrics. 

For growth performance, the final body weights of the broilers showed a strong correlation with the cumulative *C. vulgaris* intake. The sigmoid model (R^2^ = 0.616, dof = 18, *p* < 0.001) suggested a threshold effect where the body weights increased sharply up to a certain intake level before plateauing. This indicated that the optimal cumulative *C. vulgaris* intake for maximising final body weight was approximately 20 g/bird. Similarly, cumulative body weight gain followed a sigmoid pattern (R^2^ = 0.627, dof = 18, *p* < 0.001), indicating rapid weight gain up to a certain point of *C. vulgaris* intake, beyond which the gains plateaued. This suggested that the optimal cumulative intake for maximum body weight gain was also approximately 20 g/bird. Daily weight gain also showed a strong correlation with *C. vulgaris* intake, following a sigmoid model (R^2^ = 0.639, dof = 18, *p* < 0.001). This supported the threshold effect observed in other growth performance metrics, indicating that the daily weight gain optimised at a specific cumulative intake level of approximately 20 g/bird before levelling off. The FCR, a key indicator of broiler production efficiency, exhibited a lower sigmoid correlation (R^2^ = 0.289, dof = 18, *p* = 0.014), indicating that feed efficiency improved with increasing *C. vulgaris* intake up to a specific level. Beyond this level, the improvements in FCR may diminish, suggesting an optimal intake level of approximately 20 g/bird, where feed efficiency is maximised.

The intake of *C. vulgaris* in the broiler diets significantly affected various growth performances (Figure 1). Key performance indicators such as final body weight, cumulative body weight gain, and daily weight gain exhibited strong sigmoid correlations with cumulative *C. vulgaris* intake, indicating optimal intake levels of approximately 20 g/bird where these metrics were maximised. The FCR presented moderate to high correlations, suggesting specific intake levels of approximately 20 g/bird for optimal performance. These findings underscore the potential benefits and limitations of *C. vulgaris* supplementation in poultry diets, particularly concerning growth performance.

## 3. Impact of Cumulative *Chlorella vulgaris* Intake Levels on Plasma Metabolites and Immunoglobulin Levels in Broilers

Table 3 presents data from various studies examining the effects of different cumulative intake levels of *C. vulgaris* on plasma metabolites and immunoglobulin levels in broilers. The cumulative intake levels varied, providing insights into how *C. vulgaris* affects these health markers. The data showed cumulative *C. vulgaris* intake levels ranging from 1.40 g/bird [27] to 175 g/bird [16], with corresponding changes in plasma metabolites and immunoglobulin levels.

An et al. [27] investigated the effects of a 1.40 g cumulative *C. vulgaris* intake per bird over 34 days, observing total protein levels of 2.77 g/dL and triacylglycerols at 30.7 mg/dL, cholesterol at 120.3 mg/dL, high-density lipoproteins (HDL) at 97.1 mg/dL and albumin at 1.16 g/dL. The aspartate aminotransferase (AST) levels were 236.6 U/L, with plasma IgA at 721 µg/mL, IgM at 480 µg/mL and IgG at 3814 µg/mL. With a cumulative intake of 4.27 g, the final body weight was 1619 g, and the plasma metabolites showed the following slight changes: total protein at 2.78 g/dL, triacylglycerols at 34.4 mg/dL, cholesterol at 120.1 mg/dL, HDL at 93.2 mg/dL and albumin at 1.11 g/dL. The AST levels were 237 U/L, with plasma IgA at 710 µg/mL, IgM at 501 µg/mL and IgG at 3563 µg/mL. Another study [29] explored higher intake levels, such as 6.71 g and 13.04 g over 41 days. For the 6.71 g intake, the total protein was 5.050 g/dL, with triacylglycerols at 79.66 mg/dL, cholesterol at 161.00 mg/dL, HDL at 31.66 mg/dL and albumin at 2.9 g/dL. AST levels were 132.6 U/L, with plasma IgA at 290.30 µg/mL, IgM at 431.00 µg/mL and IgG at 4065.0 µg/mL. For the 13.04 g intake, the total protein was 5.600 g/dL, with triacylglycerols at 85.00 mg/dL, cholesterol at 143.30 mg/dL, HDL at 36.33 mg/dL and albumin at 3.3 g/dL. The AST levels were 97.0 U/L, with plasma IgA at 280.60 µg/mL, IgM at 328.30 µg/mL and IgG at 4600.0 µg/mL. The effects of cumulative *C. vulgaris* levels of 401 g, 561 g, and 718 g over 34 days were reported in [6]. They noted varied plasma metabolites and immunoglobulin levels across these intake levels, emphasizing the nuanced impact of high cumulative *C. vulgaris* intake on broiler health markers.

Table 4 summarises the correlation analysis for predicting the plasma metabolites and immunoglobulin levels based on cumulative *C. vulgaris* intake. Only variables with three or more degrees of freedom were analysed for correlation, ensuring the reliability of the statistical analysis. Additional data on broiler blood profiles and immune responses are provided in Appendix A. Table A1 summarises the impact of cumulative *C. vulgaris* intake levels on plasma metabolites and phitohemoglotenine-P response, while Table A2 presents the effects on the haematological profiles of the broilers.

For the systemic health indicators, the total protein levels showed a quadratic relationship with the cumulative *C. vulgaris* intake (R^2^ = 0.222, dof = 4, *p* = 0.605). The triacylglycerol levels also followed a quadratic pattern (R^2^ = 0.296, dof = 4, *p* = 0.495). The cholesterol levels showed an exponential relationship (R^2^ = 0.559, dof = 5, *p* = 0.053), suggesting that optimal cholesterol levels are achieved at lower cumulative intakes. The high-density lipoprotein levels followed a cubic pattern (R^2^ = 0.369, dof = 4, *p* = 0.399), and the albumin levels are best described by a power model (R^2^ = 0.253, dof = 4, *p* = 0.310). The AST levels showed a cubic relationship (R^2^ = 0.592, dof = 4, *p* = 0.166), suggesting that there were varying effects of *C. vulgaris* intake on liver function.

For plasma immunoglobulin levels, IgA showed a logarithmic relationship with the cumulative *C. vulgaris* intake (R^2^ = 0.569, dof = 6, *p* = 0.031), indicating that the IgA levels increased with *C. vulgaris* intake up to a point. The IgM levels followed a quadratic pattern (R^2^ = 0.746, dof = 5, *p* = 0.033), suggesting that there were optimal IgM levels at certain cumulative intakes. The IgG levels showed a cubic relationship (R^2^ = 0.842, dof = 4, *p* = 0.045), indicating that IgG levels are optimised at specific cumulative *C. vulgaris* intake levels.

The intake of *C. vulgaris* in broiler diets significantly affects plasma metabolite and plasma immunoglobulin levels. For instance, cholesterol levels are optimised at lower cumulative intakes, while IgA, IgM and IgG levels are optimised at specific intake levels, highlighting the potential health benefits of *C. vulgaris* supplementation. These findings underscore the potential benefits and limitations of *C. vulgaris* supplementation in poultry diets, particularly concerning growth performance and health markers.

## 4. Safety Precautions and Regulatory Aspects

Several critical considerations have emerged in assessing the safety precautions and regulatory aspects related to the use of *C. vulgaris* as a feed additive or ingredient in broiler diets. The safety of dietary *C. vulgaris* is generally acknowledged [34], particularly when it is free from contaminants. *C. vulgaris* is widely considered safe by regulatory authorities such as the European Food Safety Authority (EFSA) and the U.S. Food and Drug Administration (FDA), provided it is produced and processed under stringent quality control measures. Studies have shown that contaminants from freshwater sources are typically present in *C. vulgaris* at levels below detectable thresholds, reinforcing its safety profile [4]. Ensuring that *C. vulgaris* is free from contaminants like heavy metals or harmful microorganisms is essential, as these can pose significant health risks to poultry and consumers. The potential for bioaccumulation of these contaminants in broiler tissues, especially with higher levels of *C. vulgaris* intake, necessitates rigorous quality controls and regular safety assessments. This aspect underscores the need for well-established safety protocols in the production and processing of *C. vulgaris* intended for animal feed.

Proper cultivation and production conditions can ensure that the levels of contaminants such as heavy metals and harmful microorganisms remain within acceptable limits. Contamination with bacteria like *Leucobacter* sp., *Aeromicrobium* sp., *Staphylococcus* spp. and *Halomonas* spp., which can originate from various sources during the cultivation and sub-culturing processes, must be monitored and controlled. Adhering to stringent quality control measures, including regular toxicity analyses and monitoring microcystin levels, is crucial to maintaining the safety of *C. vulgaris* as a feed additive [5].

Additionally, the regulatory landscape surrounding the use of *C. vulgaris* in animal feed is complex and varies across different regions. Compliance with local and international regulations concerning feed safety, permissible additive levels, and labelling requirements is paramount. These regulations have been designed to ensure the safety of animal feed additives and, by extension, the safety of animal-derived food products for human consumption. Regulatory standards often evolve in response to new scientific findings and public health considerations. For instance, studies contribute to a growing body of evidence that regulators may use to review and update the guidelines on the use of *C. vulgaris* in poultry diets [4].

The long-term safety of *C. vulgaris*, particularly at high inclusion levels and over extended feeding durations, remains an area requiring further research. While short-term studies have indicated beneficial effects, the long-term implications for animal and human health are not fully understood. This knowledge gap calls for ongoing research and monitoring to detect any potential adverse effects, including the cumulative impacts of bioactive compounds in *C. vulgaris* on animal health and food safety.

In summary, while *C. vulgaris* offers potential health benefits as a poultry feed additive, its safe inclusion in broiler diets demands a comprehensive approach encompassing rigorous quality control, adherence to evolving regulatory standards and continuous research into its long-term safety and efficacy. Such an approach is essential to ensure that *C. vulgaris*-enhanced broiler meat is not only beneficial but also safe and compliant with regulatory requirements, thereby maintaining consumer trust and market viability.

## 5. Conclusions and Future Research

The intake of *C. vulgaris* in broiler diets significantly influences broiler growth performance and health-related compounds. Key performance indicators such as final body weight, cumulative body weight gain and daily weight gain exhibit significant sigmoid correlations with cumulative *C. vulgaris* intake, with optimal intake levels at approximately 20 g/bird, where these metrics are maximised. However, the feed conversion ratio presents lower significant sigmoid correlations, which also suggests specific intake levels of approximately 20 g/bird for optimal performance.

For health-related compounds, plasma metabolites and immunoglobulin levels demonstrate significant correlations with cumulative *C. vulgaris* intake. For instance, the levels of total protein, triacylglycerols and cholesterol have shown varying relationships with *C. vulgaris* intake, indicating that optimal intake levels can help maintain balanced plasma metabolites. Plasma immunoglobulin levels, particularly IgA, IgM and IgG, have exhibited significant correlations with *C. vulgaris* intake, suggesting enhanced immune responses at specific intake levels. Higher levels of immunoglobulins inside the normal range, particularly IgM and IgG, are generally indicative of an enhanced immune response. Increased IgM levels typically suggest a primary immune response, indicating that the bird’s immune system is effectively recognizing and responding to antigens. Elevated IgG levels are associated with long-term immunity and memory response, suggesting that the birds are better prepared to fight off infections.

These findings underscore the potential benefits of *C. vulgaris* supplementation in broiler diets. Optimal cumulative intake levels of approximately 20 g/bird maximise growth performance and positively influence health markers, including plasma metabolites and immunoglobulin levels. However, it is crucial to balance these benefits with potential diminishing returns or adverse effects at higher intake levels, ensuring that *C. vulgaris* is used effectively within the dietary framework of broilers.

Future research on *C. vulgaris* in poultry nutrition should concentrate on identifying the optimal dosage and duration of supplementation to maximise growth performance and health benefits while avoiding adverse effects. Longitudinal studies are essential to evaluate the long-term implications of *C. vulgaris* use on broiler health, particularly concerning the potential accumulation of bioactive compounds. Additionally, understanding the mechanisms by which *C. vulgaris* influences broiler physiology will help refine supplementation strategies for targeted outcomes, ensuring that broilers receive the maximum benefit from *C. vulgaris*’s nutritional properties.

Moreover, ensuring the safety and regulatory compliance of *C. vulgaris* supplementation is crucial. Rigorous quality control measures must be implemented to prevent contamination risks, and adherence to regulatory guidelines is necessary to maintain consumer confidence in *C. vulgaris*-supplemented poultry products. Comparative studies with other feed additives could also provide valuable insights into *C. vulgaris*’s relative effectiveness and economic viability, helping producers make informed decisions about its use in poultry diets. By addressing these key areas, future research can enhance our understanding of how to optimise *C. vulgaris* supplementation, ultimately improving both the productivity and health of broilers. 

## Figures and Tables

**Figure 1 vetsci-11-00290-f001:**
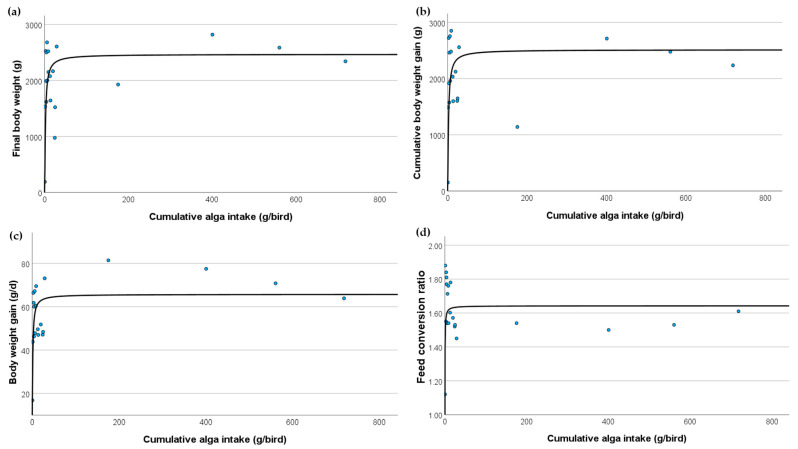
The relationship between *Chlorella vulgaris* dosage and key broiler growth indicators characterized by sigmoid models: (**a**) final body weight, (**b**) cumulative body weight gain, (**c**) body weight gain, and (**d**) feed conversion ratio.

**Table 2 vetsci-11-00290-t002:** Summary of the correlation analysis for predicting the dependent performance variables based on cumulative *Chlorella vulgaris* intake.

Variable	Best Model Type	R-Square	Degrees of Freedom	*p*-Value	Model Equation
Final body weight (g)	Sigmoid	0.616	18	<0.001	y = 7.811/(1 + e^−1.550 × (x − x0)^)
Cumulative body weight gain (g)	Sigmoid	0.627	18	<0.001	y = 7.830/(1 + e^−1.685 × (x − x0)^)
Body weight gain (g/d)	Sigmoid	0.639	18	<0.001	y = 4.185/(1 + e^−0.905 × (x − x0)^)
Feed conversion ratio	Sigmoid	0.131	18	0.117	y = 0.496/(1 + e^−0.138 × (x − x0)^)

**Table 3 vetsci-11-00290-t003:** Impact of cumulative *Chlorella vulgaris* intake levels on plasma metabolites and immunoglobulin levels in broilers.

Initial Age and Weight	Alga Level (% Feed) and Duration of Trial (Days) ^1^	Cumulative Alga Intake (g/bird) ^2^	Plasma Metabolites						Plasma Immunoglobulins			Reference
			Total Protein (g/dL)	Triacylglycerols (mg/dL)	Cholesterol (mg/dL)	HDL (mg/dL)	Albumin (g/dL)	AST (U/L)	IgA (µg/mL)	IgM (µg/mL)	IgG (µg/mL)	
45.1 g, 1 d-old ^3^	0.05%, 34 d	1.40	2.77	30.7	120.3	97.1	1.16	236.6	721	480	3814	[27]
45.1 g, 1 d-old ^3^	0.15%, 34 d	4.27	2.78	34.4	120.1	93.2	1.11	237.0	710	501	3563	[27]
41.8 g, 1 d-old	0.20%, 41 d	6.71	5.050	79.66	161.0	31.66	2.90	132.6	290.3	431.0	4065	[29]
41.8 g, 1 d-old	0.40%, 41 d	13.0	5.600	85.00	143.3	36.33	3.30	97.00	280.6	328.3	4600	[29]
45.1 g, 1 d-old ^3^	0.50%, 34 d	14.1	2.72	34.1	109.5	86.1	1.09	242.5	602	322	3827	[27]
41.8 g, 1 d-old	0.60%, 41 d	20.0	4.300	74.00	168.6	28.33	2.50	71.00	282.3	454.0	5613	[29]
1 d-old ^3^	1.0%, 34 d	24.4	-	-	-	-	-	-	291.37	70.95	645.15	[32]
1 d-old ^3,4^	1.0%, 34 d	25.2	-	-	-	-	-	-	291.95	70.58	686.40	[32]
788.3 g, 21 d-old ^3^	10%, 14 d	175	2.893	40.2	79.9	56.6	-	297.2	-	-	-	[16]

^1^ The final day of the trial, which involved slaughtering, was not included in the trial’s duration. ^2^ Calculated by multiplying the total feed consumed per animal during the experimental period by the dietary percentage of microalga. For some of the studies, no information about the cumulative feed intake was available, and therefore, an estimation of this was completed as follows: cumulative feed intake (g/bird) (An et al. [27]) = cumulative feed intake (g/d/bird) × number of trial days and cumulative feed intake (g/bird) (Coelho et al. [16]) = cumulative feed intake (g/pen)/number of birds. ^3^ Male broilers. ^4^ In this trial, fresh liquid *Chlorella vulgaris* was used. AST, aspartate aminotransferase (EC. 2.6.1.1); HDL, high-density lipoproteins.

**Table 4 vetsci-11-00290-t004:** Summary of correlation analysis for predicting the dependent variables, plasma metabolites and immunoglobulins, based on the cumulative *Chlorella vulgaris* intake.

Variable	Best Model Type	R-Square	Degrees of Freedom	*p*-Value	Model Equation
Total protein (g/dL)	Quadratic	0.222	4	0.605	y = 3.143 + 0.080x
Triacylglycerols (mg/dL)	Quadratic	0.296	4	0.495	y = 37.286 + 2.083x − 0.012x^2^
Cholesterol (mg/dL)	Exponential	0.559	5	0.053	y = 139.056 × e^−0.003x^
HDL (mg/dL)	Cubic	0.369	4	0.399	y = 91.784 − 3.232x + 0.017x^2^ + 0.000x^3^
Albumin (g/dL)	Power	0.253	4	0.310	y = 1.053 × x^0.269^
AST (U/L)	Cubic	0.592	4	0.166	y = 240.834 − 7.247x (b2 and b3 are 0)
IgA (µg/mL)	Logarithmic	0.569	6	0.031	y = 786.178 − 153.886 × log(x)
IgM (µg/mL)	Quadratic	0.746	5	0.033	y = 463.123 + 4.418x − 0.740x^2^
IgG (µg/mL)	Cubic	0.842	4	0.045	y = 4805.081 − 614.095x + 77.663x^2^ − 2.389x^3^

AST, aspartate aminotransferase (EC. 2.6.1.1); HDL, high-density lipoproteins.

## Data Availability

The data presented in this study are available on request from the corresponding author.

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
