# Peer review of "Impact of Chlorella vulgaris Intake Levels on Performance Parameters and Blood Health Markers in Broiler Chickens"

_vetsci, 2024, doi:10.3390/vetsci11070290_

Round 1
Reviewer 1 Report
Comments and Suggestions for Authors
This review highlights the benefits of adding chlorella to broiler diets, where an optimal cumulative intake of around 20 grams per chicken not only maximizes growth performance, but also has a positive impact on health indicators.
It is also critical to ensure the safety and compliance of chlorella additives. It is necessary to establish the quality standard of chlorella additives and to implement strict quality control measures to prevent the risk of contamination. Comply with regulatory guidelines to ensure quality standards for chlorella additions.
Several experiments involved in this paper are using fresh liquid chlorella. Is the proportion of fresh liquid chlorella added to the feed the proportion of dry matter of fresh liquid Chlorella?
Author Response
Reviewer 1
This review highlights the benefits of adding Chlorella to broiler diets, where an optimal cumulative intake of around 20 grams per chicken not only maximizes growth performance, but also has a positive impact on health indicators.
Reply: Thank you for your comments and suggestions. We appreciate it and we will try to address all of them.
It is also critical to ensure the safety and compliance of Chlorella additives. It is necessary to establish the quality standard of Chlorella additives and to implement strict quality control measures to prevent the risk of contamination. Comply with regulatory guidelines to ensure quality standards for Chlorella additions.
Reply: Thank you for your comments. We agree that ensuring the safety and regulatory compliance of Chlorella additives is crucial. Chapter 4, "Safety Precautions and Regulatory Aspects," comprehensively addresses the need to establish quality standards for Chlorella additives and implement strict quality control measures to prevent contamination risks. This chapter also emphasises the importance of adhering to regulatory guidelines to ensure the safe application of Chlorella in animal feed. We will ensure that this section thoroughly reflects these critical considerations.
Several experiments involved in this paper are using fresh liquid Chlorella. Is the proportion of fresh liquid Chlorella added to the feed the proportion of dry matter of fresh liquid Chlorella?
Reply: Thank you for your comment. We acknowledge that only one treatment in the study by Kang et al. [30] used fresh liquid Chlorella vulgaris. The proportion of fresh liquid Chlorella added to the feed is not equivalent to the proportion of dry matter. Unfortunately, the study does not provide sufficient information to convert the fresh liquid Chlorella proportion to its dry matter equivalent.
Reviewer 2 Report
Comments and Suggestions for Authors
There are only two observations:
- I think the simple summary provided could be even more simplified to enhance clarity and conciseness.
- While the current literature referenced is recent, it may benefit from a broader scope of sources if more sufficient research is available.
Author Response
Reviewer 2
I think the simple summary provided could be even more simplified to enhance clarity and conciseness.
Reply: Thank you for your suggestion. We appreciate your feedback on enhancing the clarity and conciseness of the simple summary. We will revise the summary to make it more straightforward and easier to understand, ensuring that the key points are clearly communicated to the reader.
While the current literature referenced is recent, it may benefit from a broader scope of sources if more sufficient research is available.
Reply: Thank you for your suggestion. We have conducted an extensive literature search and, unfortunately, there are no additional scientific data available on this topic. We have included all relevant and recent sources to provide a comprehensive review.
Reviewer 3 Report
Comments and Suggestions for Authors
The manuscript titled "Impact of Chlorella vulgaris Intake Levels on Performance Parameters and Blood Health Markers in Broiler Chickens" was well written and had no major structure problems. The abstract was concise and allowed the reader to have a general idea of the scientific content. The main goal of the review was clearly stated, the statistical methods used were appropriate.
I have only a few suggestions for improvement, and they are mainly related to providing a little more background information to support the significant findings. For example:
1. Starting on line 81, the authors mentioned that Chlorella supplementation increases the levels of certain immunoglobulins and considered this effect beneficial. Then, in the body of the review, there was a good explanation of the relationships found - the effect of supplementation on the behavior of the immunoglobulins. For example, starting on line 339, the authors stated: "suggesting enhanced immune responses at specific intake levels". But, the authors never explained what is the physiological meaning of having higher levels of these immunoglobulins. I could argue that increased levels of IgM and IgG might indicate the bird is fighting a pathogen (or something the immune systems is considering foreign). In summary, my question is "What makes you say that higher immunoglobulins levels is a good thing?"
2. In several places throughout the narrative points of comparison were missing. For example, the paragraph starting on line 132 provided several examples of growth performance data for birds supplemented with Chlorella, however, it never mentioned the data for the control birds so readers could have a point of comparison. It might be helpful to add that information. I assumed the data was statistically significant, but maybe adding some of the control could help. This is just a suggestion.
In general, this manuscript should be published as it provides important information to the industry.
Author Response
Reviewer 3
The manuscript titled "Impact of Chlorella vulgaris Intake Levels on Performance Parameters and Blood Health Markers in Broiler Chickens" was well written and had no major structure problems. The abstract was concise and allowed the reader to have a general idea of the scientific content. The main goal of the review was clearly stated, the statistical methods used were appropriate.
Reply: Thank you for your positive feedback on our manuscript. We are pleased to hear that you found the structure well-organized and the abstract concise and informative. We appreciate your acknowledgement of the clarity of our main goal and the appropriateness of our statistical methods. Your comments have been carefully considered, and we will continue to ensure that our work maintains high standards of clarity and scientific rigour.
I have only a few suggestions for improvement, and they are mainly related to providing a little more background information to support the significant findings. For example:
- Starting on line 81, the authors mentioned that Chlorella supplementation increases the levels of certain immunoglobulins and considered this effect beneficial. Then, in the body of the review, there was a good explanation of the relationships found - the effect of supplementation on the behavior of the immunoglobulins. For example, starting on line 339, the authors stated: "suggesting enhanced immune responses at specific intake levels". But, the authors never explained what is the physiological meaning of having higher levels of these immunoglobulins. I could argue that increased levels of IgM and IgG might indicate the bird is fighting a pathogen (or something the immune systems is considering foreign). In summary, my question is "What makes you say that higher immunoglobulins levels is a good thing?"
Reply: Thank you for your valuable feedback. We appreciate your suggestion to provide more background information to support our findings on immunoglobulin levels. Higher levels of immunoglobulins inside the normal range, particularly IgM and IgG, are generally indicative of an enhanced immune response. Increased IgM levels typically suggest a primary immune response, indicating that the bird's immune system is effectively recognizing and responding to antigens. Elevated IgG levels are associated with long-term immunity and memory response, suggesting that the birds are better prepared to fight off infections. These immunoglobulins play crucial roles in pathogen neutralization, opsonization, and complement activation, thereby enhancing the overall immune defence mechanism. Therefore, the observed increase in immunoglobulin levels due to Chlorella supplementation is considered beneficial as it implies a more robust and responsive immune system in broiler chickens. We will include this explanation to provide a clearer understanding of the physiological significance of our findings.
- In several places throughout the narrative points of comparison were missing. For example, the paragraph starting on line 132 provided several examples of growth performance data for birds supplemented with Chlorella, however, it never mentioned the data for the control birds so readers could have a point of comparison. It might be helpful to add that information. I assumed the data was statistically significant, but maybe adding some of the control could help. This is just a suggestion.
Reply: Thank you for your suggestion. We have added the control data to the relevant sections to provide a clear point of comparison, specifically in lines 132 to 150. This addition will help readers better understand the significance of the growth performance data for birds supplemented with Chlorella vulgaris.
In general, this manuscript should be published as it provides important information to the industry.
Reply: Thank you for your comments and suggestions. We appreciate your support and are pleased to hear that you find our manuscript valuable to the industry. We have taken your feedback into account to improve the clarity and comprehensiveness of our work.
Reviewer 4 Report
Comments and Suggestions for Authors
The review establishes a clear relationship between cumulative Chlorella intake and significant outcomes in poultry nutrition and health through a detailed analysis of various studies. However, there are still some issues that need to be determined.
1. Lines 26-28: The statement is incorrect.
2. When to use the full name “Chlorella vulgaris” and when to use the abbreviation “Chlorella” in the text?
3. Did the authors show that 20g per chicken is the optimal intake, and there is no correlation with the age of the chickens?
4. Lines 143-146: What does this sentence mean?
5. The authors describe a narrative of previous studies (Table 1), but the manuscript does not summarize the commonalities and differences, and what the implications of these studies are.
6. The authors describe previous studies (Table 3) and again do not see the authors' insights.
7. This review does not mention new perspectives on the topic.
Author Response
Reviewer 4
The review establishes a clear relationship between cumulative Chlorella intake and significant outcomes in poultry nutrition and health through a detailed analysis of various studies. However, there are still some issues that need to be determined.
Reply: Thank you for your feedback. We appreciate your recognition of our work in establishing the relationship between cumulative Chlorella intake and its effects on poultry nutrition and health. We have carefully considered all your comments and suggestions to address the issues raised and improve the manuscript accordingly.
- Lines 26-28: The statement is incorrect.
Reply: Thank you for pointing that out. We have corrected the statement in lines 26-28 to ensure accuracy.
- When to use the full name “Chlorella vulgaris” and when to use the abbreviation “Chlorella” in the text?
Reply: Thank you for your comment. We have ensured that the full name "Chlorella vulgaris" is used the first time it appears in the text, followed by the abbreviation "C. vulgaris" in subsequent mentions, as per conventional standards.
- Did the authors show that 20 g per chicken is the optimal intake, and there is no correlation with the age of the chickens?
Reply: Thank you for your comments. Our work aims to clarify the optimal intake level of Chlorella in chickens. We reviewed a range of studies where Chlorella intake varied from 2.95 g/bird to 20 g/bird, all conducted with 1-day-old chicks over trials lasting 41 days [27, 28, 29]. Based on the available data, 20 g per bird appears to be optimal for growth performance. Regarding the correlation with chicken age, our review primarily focused on cumulative intake levels and growth outcomes. Specific correlations with age were not extensively documented in the studies reviewed. More research is needed to establish clearer links between optimal intake, growth performance, and the age of the chickens, as the current literature does not provide a definitive answer on this aspect.
- Lines 143-146: What does this sentence mean?
Reply: Thank you for your comment. We revised the sentence for clarification.
- The authors describe a narrative of previous studies (Table 1), but the manuscript does not summarize the commonalities and differences, and what the implications of these studies are.
- The authors describe previous studies (Table 3) and again do not see the authors' insights.
Reply: Thank you for your comment. This review aims to synthesize existing information rather than generate new findings. Our objective was to describe and draw conclusions based on a compilation of data from various previous studies. We have incorporated multiple correlation analyses to objectively derive insights from the data, highlighting commonalities, differences and implications. However, future research is necessary to further explore and validate these findings.
- The authors describe previous studies (Table 3) and again do not see the authors' insights.
Reply: Thank you for your comments. This follows the same reasoning as the previous question. We have incorporated multiple correlation analyses to objectively derive insights from the data, highlighting commonalities, differences and implications. However, future research is necessary to further explore and validate these findings.
- This review does not mention new perspectives on the topic.
Reply: Thank you for your comment. We believe our paper contributes novel perspectives by providing a comprehensive analysis of cumulative microalgae Chlorella vulgaris intake. We employ model equations for precise correlation descriptions and identify saturation points in broiler responses. Additionally, by synthesising existing literature, we offer a thorough overview that advances understanding in the field of poultry nutrition and supplementation with Chlorella vulgaris. These aspects collectively pave the way for informed dietary strategies and further scientific exploration.
Round 2
Reviewer 4 Report
Comments and Suggestions for Authors
Thank you very much!